# Cradle-to-Gate Life Cycle Assessment (LCA) of GaN Power Semiconductor Device

**Laura Vauche \*, Gabin Guillemaud, Joao-Carlos Lopes Barbosa and Léa Di Cioccio**

CEA, LETI, University Grenoble Alpes, F-38000 Grenoble, France; joao-carlos.lopesbarbosa@cea.fr (J.-C.L.B.);
lea.dicioccio@cea.fr (L.D.C.)
* Correspondence: laura.vauche@cea.fr

**Abstract:** Wide Band Gap (WBG) semiconductors have the potential to provide significant improvements in energy efficiency over conventional silicon (Si) semiconductors. While the potential for energy efficiency gains is widely researched, the relation to the energy and resource use during manufacturing processes remains insufficiently studied. In order to appraise the performance of the technology thoroughly, issues such as raw material scarcity, toxicity and environmental impacts need to be investigated in detail. However, sparse Life Cycle Assessment (LCA) data are available for the two currently most widespread WBG semiconductor materials, gallium nitride (GaN or GaN/Si) and silicon carbide (SiC). This paper, for the first time, presents a cradle-to-gate life cycle assessment for a GaN/Si power device. To allow for a full range of indicators, life cycle assessment method EF 3.1 was used to analyze the results. The results identify environmental hotspots associated with different materials and processes: electricity consumption for the processes and clean room facilities, direct emissions of greenhouse gases, gold (when used), and volatile organic chemicals. Finally, we compare this result with publicly available data for Si, GaN and SiC power devices.

**Keywords:** GaN; power devices; environmental impact assessment; cradle-to-gate; life cycle assessment

## 1. Introduction

Rapidly transitioning the world away from the use of fossil fuels to cleaner renewable forms of energy is essential if the world is to meet climate targets. Part of greenhouse gas emissions reduction could be achieved with the use of renewable energy and a high rate of electrification instead of fossil fuels. The broader application of wide bandgap (WBG) semiconductors for power electronics carries the promise of large energy savings in a range of different applications, including inverters for photovoltaic systems, power supplies for consumer electronics, uninterrupted power supplies for data centers and drive-trains and charging infrastructure for the electric automotive sector [1,2]. WBG semiconductors are characterized by bandgap energies three or more times that of Si, enabling them to withstand much higher voltages. Consequently, WBG devices can be made much smaller, allowing faster switching with less resistance. Because of the reduced resistance, less energy is wasted as heat compared with Si devices, making WBG more energy efficient. While these energy efficiency improvements from using WBG components compared with Si-components are widely researched and promoted by manufacturers, the environmental impacts along the entire life cycle (beyond the use phase) are far less understood. Moreover, to the best knowledge of the authors, there is currently no Life Cycle Assessment (LCA) data available specifically focusing on WBG semiconductors, and (publicly) available LCA information of sufficient level of detail and quality is limited or outdated [3,4]. This is due to process complexity involving numerous process steps and specialty chemicals in energy-intensive clean rooms, and rapidly evolving technology in a competitive and secretive industry. In addition, most of the current research available in the field of semiconductors focuses rather on Si-based semiconductors for information technology or solar cells and



does not take into account specificities of power semiconductors such as wafer thickness, specific processing, number of mask levels, etc. In this paper, we evaluate the environmental impacts at the fabrication of a GaN/Si transistor, of specifications 650 V and 30 A [5], on a CMOS-compatible process flow on 200 mm Si substrate, with LCA methodology.

## 2. Methodology and Scope

Life cycle assessment (LCA) is a method that helps investigate the environmental impact of a product from raw material extraction to disposal or recycling. This is achieved by quantifying all materials and energy inputs as well as waste and pollutants outputs, enabling the comparison of the impacts of products manufactured and used for the same purpose. The present LCA study was performed according to the international standards, ISO 14040 and ISO 14044 [6,7], using SimaPro software with the ecoinvent v3.9 database. The EF 3.1 methodology was applied to analyze the results, as recommended by the European Commission.

The LCA methodology outlines four stages: (1) goal and scope definition, (2) inventory analysis, (3) impact assessment, (4) interpretation and, if possible, sensitivity analysis. The LCA performed in this study was carried out considering all phases from raw material extraction to the production gate, following a cradle-to-gate assessment method.

### 2.1. Goal and Scope Definition

This study aims to assess the environmental impacts of producing a GaN semiconductor power device. Since the normally-off GaN devices offer the best cost/benefit from an application point of view (lower switching and conduction losses) [8], this study focuses on the normally-off GaN metal-oxide-semiconductor (MOS) channel high-electron-mobility transistor (HEMT), also called MOSc-HEMT, developed at CEA-LETI [5]. Inventory data for the production was obtained from in-house CEA-LETI R&D microelectronics clean rooms, where GaN on Si semiconductor technology is developed on 200 mm silicon wafers with a CMOS compatible process flow, with 2021 data.

In this study, the LCA was performed for the following functional unit: To produce a transistor which allows the flow of a drain ON-current of 30 A, with a drain-to-source blocking voltage capability of 650 V, which corresponds to the electrical specifications of one power transistor. It should be noted that a semiconductor device alone does not provide functionality, but must be part of a larger system such as a converter [9], whose design depends on the power device electrical performance and application, which can range from solar microinverters, servers, telecom & datacom equipment, adapters/chargers, wireless charging and audio, to vehicle electrification. An example of a converter functional unit is: "Generate a three-phase AC electrical operating point for a 150 kW load (electric machine) from a 450 V DC source, based on a lifespan of 15 years, i.e., equivalent to 10,000 h of operation" [10].

Following this analysis, a "cradle-to-gate" approach, which only takes into account impacts occurring from raw material extraction to the production gate, was preferred, as the use phase depends on the chosen system and its technical performance, Figure 1.

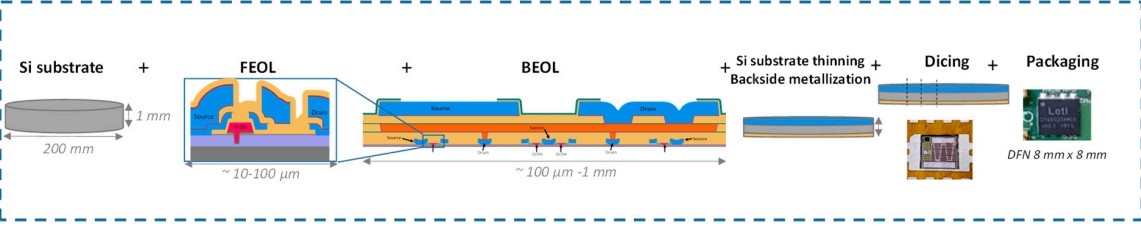

**Figure 1.** Drawing showing the system boundary detailing the stages included into this cradle-to-gate analysis. It only consists of the manufacturing stage of device production, starting with raw material extraction, through processes for the production of a wafer, followed by dicing and packaging of the individual devices ("dies") into transistor chips.

### 2.2. Life Cycle Inventory (LCI)

Inventory data have been collected, where available, in the cleanroom. Missing data were modeled, based on data collected from patents, publications or industrial data sheets. An hybrid approach was chosen, imaged in Figure 2 and described hereafter.

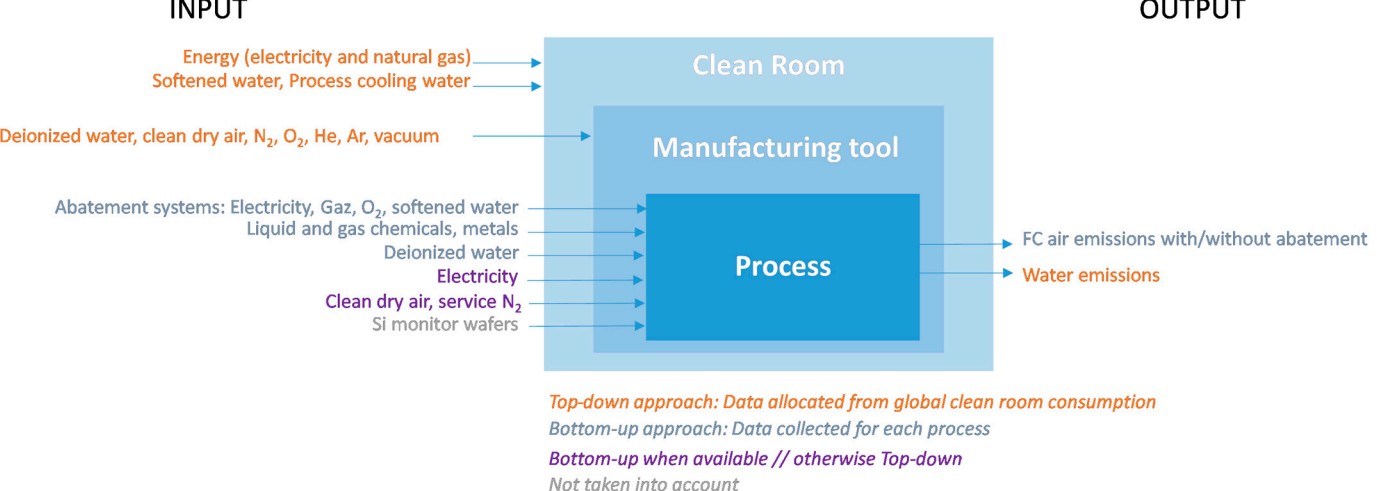

**Figure 2.** Shows the hybrid top-down and bottom-up approach used for semiconductor device manufacturing LCI in this study.

Substrate production. Si and GaN/Si semiconductors are manufactured using high-purity industrial-grade silicon, by growing Si ingots through the Czrochalski process with typical diameters of 8 to 12 inches (200 to 300 mm). In this study, the GaN power device is manufactured using 200 mm Si substrates. Czrochalski Si wafers of 200 mm diameter have been modelled using the LCI of the photovoltaic crystalline silicon module supply chain (polycrystalline silicon feedstock purification, crystallization, wafering) [11], adapted for 1 mm thick substrates produced in China.

Wafer processing. Semiconductor devices are among the most complicated of all manufactured products: the same production line can be used for hundreds of different products, and production tools can be used many times for the same product. Wafer processing strongly depends on the complexity of the final chip. Integrated circuits (ICs) like semiconductor memories, processors or logic chips are at the end of the complexity spectrum, often requiring hundreds of wafer processing steps, with a range of different high-purity chemicals applied to create layers or so called "mask levels". Semiconductor manufacturing consists of two major stages of front-end-of-line (FEOL) for the active part of the device (gate, source, drain), and back-end-of-line (BEOL) for the metallic connections. The processing steps are classified by process category, such as:

- Epitaxy;
- Deposition of thin films by physical or chemical processes, e.g., CVD (chemical vapor deposition), PVD (physical vapor deposition), PECVD (plasma-enhanced chemical vapor deposition), ALD (atomic layer deposition);
- Photolithography, in which a pattern is transferred from a mask to a sacrificial photo-sensitive material (photoresist);
- Dry etching, resulting in the transfer of the pattern to the thin films;
- Photoresist stripping;
- Wet chemical wafer cleaning (wet bench or spray cleaning);
- Chemical–mechanical planarization (CMP).

Metrology and characterization steps are present in order to assess the conformity of processing and to characterize the physico-chemical or electrical properties of the different layers and interfaces at different stages of the processing. Some of these steps are specific

for R&D purposes and would not be present in an industrial process flow. In this study, all metrology and characterization steps were neglected, as their impacts are very small compared to those of the process steps. The consumption of Si monitor wafers as holders or as dummies, and the manufacturing of tools and construction of the clean room buildings, were not taken into account in this study, as it was not possible to get accurate information on their impacts. The sixteen impact factors were analyzed.

Working in a clean room environment contributes to large electricity, natural gas and water consumption for climate and humidity control, with overpressure and particle filtration, extensive subfab facilities for gas abatement, exhaust pumps, water chillers, and water purification. Therefore, a specific methodology is needed for the LCA of a semiconductor device, taking into account processing, maintaining the manufacturing tools ready for process (also called "idle mode") and maintaining a clean room infrastructure [12]. An hybrid top-down and bottom-up approach was used for this life cycle inventory (LCI) [13,14].

A bottom-up approach was used for the processes, i.e., consumptions of chemicals, gases, metals and deionized or ultra-pure water (UPW) were collected individually for each process step. For wet benches, the volume of chemical used was taken over an extended period of time and divided by the number of wafers treated in the chemistry tank. The consumption for test runs (which might be required before every production run) was taken into account. The consumption of non-production runs made on a "per unit of time" basis was distributed across the number of wafers produced during that time period [15]. When applicable, batch processing was considered for simultaneous treatment of 12 wafers. When possible, electricity consumption was directly measured during the processing.

Emissions to air were estimated for dry-etching or deposition chamber cleaning processes emitting fluorinated compounds (FCs). FCs are an important group of greenhouse gas emissions due to their high infrared absorption and long lifetimes. The proportion of FC gases not used in the process and the formation of FC by-products were calculated using IPCC Tier2b and Tier2c 200 mm default emission factors [16]. When a combustion and water scrubbing abatement system was present, the emissions to air were calculated after the abatement system into emissions to air and to water using default destruction removal efficiency (DRE) parameters for each FC [16], and the consumption of softened water, natural gas, electricity and $O_2$ of the abatement system itself was added according to the manufacturer datasheet and the duration of the process.

A top-down approach was used in order to allocate global consumption and emissions to each process step according to allocation rules proposed by Lopes Barbosa et al. [13].

(i)   Electricity consumption was preferentially estimated from direct power measurements carried out using a standard recipe, when available (bottom-up approach). When these data were not accessible, the consumptions of clean dry air (CDA), service nitrogen and electricity during the processing were allocated to each process step according to the ratio of tool surface/clean room surface and wafer-second processing time. The top-down allocation can lead to an underestimation of the electricity consumption for processes with plasma or requiring heating (see Table 1).

**Table 1.** Ratio between electricity consumption measured during standard processing and electricity consumption calculated from global clean room consumption.

| Tool Process Category | Ratio of Electricity Consumption from Direct Power Measurement (Clamp-On Amp Meter)/Electricity from Top-Down Allocation |
|---|---|
| Dry etching | 4.7 |
| Stripping | 3.1 |
| PECVD deposition | 3.0 |
| Epitaxy | 2.4 |
| Photolithography | 0.9 |

(ii) The continuous consumptions by the clean room grids of nitrogen, argon, oxygen, helium, ultra-pure deionized water and house vacuum (i.e., water and electricity to maintain vacuum), in order to maintain manufacturing tools ready for the process (also called "idle mode") were allocated to each process step according to the number of tools in the clean room connected to a specific grid and wafer-second processing time.

(iii) The clean room facility consumptions of electricity (excluding electricity for the processes), natural gas, softened water and process cooling water (PCW) were allocated to each process step per ratio of tool surface/clean room surface and wafer-second processing time.

(iv) The liquid waste treatment was allocated according to $m^3$ of liquid waste directed into a specific drain (acido-basic effluent drain, fluorhydric drain, solvent drain or solvent collection). The different treatments, such as addition of HCl or NaOH to neutralize the pH of acido-basic effluents, addition of flocculent material to produce non-hazardous solids containing calcium fluorite out of fluorhydric drain, or incineration, were taken into account separately for each specific drain based on publicly available information and internal CEA data [17].

The dataset was taken from the ecoinvent database in order of preference market for France {FR}, Europe {RER} and global {GLO}. When the composition of a proprietary chemical was not available, it was modelled with an average composition. It should be noted that when LCA data were available in the ecoinvent database for elemental gases, metals and common chemicals, they were usually representative of the industrial grade, with a purity of 99% or lower, rather than ultra-high-purity or semiconductor grade. When data were not present in the database, they were modelled in order of preference according to life cycle inventories available in literature: $NF_3$ [18], TMGa [19], TMAl [19]:

(i) With the reactants necessary for their chemical synthesis, according to the methodology described by Huber et al. [20]: stripping chemistries and photosensitive materials;

(ii) Replaced by a proxy or an isomer available in the ecoinvent database: CHF3 by CH2F2, 2 2-aminoethoxyethanol (N°CAS: 000929-06-6) by diethanolamine;

(iii) Modelled only with associated energy intensity (MJ/kg) and carbon intensity ($CO_2$ eq/kg) used by Boyd et al. (Tables 4.2–4.5) [21]: DCS, $C_4F_8$, $CF_4$.

Packaging. After wafer-level processing (including wafer thinning and backside metallization), each wafer contains hundreds of individual devices called dies, which are tested, diced and packaged into chips. Chips are enclosed inside a protective package with external leads or connectors. There are many families of packages. Due to the standardization of semiconductor packaging, the adoption of a new packaging type is a slow process, which also depends on the devices' prevalence. As GaN devices are not yet widely common, they are usually packaged in the same type of package as Si-based devices. A common packaging is lead frame packaging technology to connect a die to a printed circuit board (PCB) such as quad flat no-lead package (QFN). In this study, for the MOSc-HEMT, the packaging is outsourced. Processes for the packaging of dies, transportation, mounting and assembly of packaged chips were therefore excluded from this study. Despite the long distances that semiconductor wafers and chips are typically shipped during production and prior to use, the impacts of transportation are almost insignificant due to the small mass of the product [22]. A bill-of-materials approach was used to determine the amounts of materials from the package dimensions.

### 2.3. Life Cycle Assessment

For this study, the authors used the LCA methodology based on the ISO 14040 standard.

The EF 3.1 midpoint methodology was applied to analyze the results, as recommended by the European Commission. Sixteen impact categories were selected according to the EF 3.1 method. These categories included climate change; ozone depletion; ionizing radiation; photochemical ozone formation; particulate matter; human toxicity, non-cancer; human toxicity, cancer; acidification; eutrophication, freshwater; eutrophication, marine;

eutrophication, terrestrial; ecotoxicity, freshwater; land use; water use; resource use, fossils; and resource use, minerals and metals. In the classification and characterization stage, the impact magnitude of all inputs and outputs collected in the inventory is quantified for each impact category by multiplying the physical quantities by the characterization factors provided by the models. Each impact category has an environmental score in a different unit, for instance, kg $CO_2$ eq for the climate change. After characterization, the results are then normalized by a reference value (per capita impacts of an average person in Europe over one year) and pondered with weighting factors for each impact category. The resulting scores can be added together to provide a single score. The overall single score is dimensionless, measured in "points" or Pt. The interpretation and hotspot analysis are carried out on the most relevant impact categories which contribute cumulatively to at least 80% of the total environmental impact expressed as single score.

### 2.4. Sensitivity Analysis

Production yield, die size, geographical location and the choice to abate fluorinated compounds have been previously identified as the most important metrics impacting energy and global warming potential impacts in the production stage, and will be discussed [22].

## 3. LCI, LCA Results

### 3.1. Life Cycle Inventory for the Production of one 200 mm Wafer, Processed with GaN on Si MOSc-HEMT Technology

In this study, the GaN MOSc-HEMT processing requires 17 mask levels (10 for the front-end processing or FEOL, six for the back-end processing or BEOL, and one for wafer substrate thinning, which is specific to the chosen packaging) and 118 processing steps. Power semiconductors like MOSFETs and diodes require far fewer mask levels compared with memories, processors or logic chips, e.g., SiC IGBTs require about 11 to 12 mask levels [23]. FEOL requires 56 process steps; BEOL, 50 process steps; and Si wafer thinning, 12 process steps. For a 200 mm wafer, after processing and Si substrate thinning, the material amount present on the wafer is between 6 and 67 times lower than what was necessary for the processing, as shown in Table 2. The amount of material that must be extracted to produce a certain amount of metal commodity is even greater, with a rock-to-metal ratio from 3 for Si to $3 \times 10^6$ for Au [24].

**Table 2.** Shows the amount of various elements present on the wafer and the amount that is consumed (elemental, as calculated from gas or liquid precursor quantity) during wafer processing. * Gold is recovered from the deposition chamber.

| Element | Amount Present on the Processed Wafer (g) | Amount Consumed during Wafer Processing (g) | Factor |
|---|---|---|---|
| Si | 12.066 | 77.650 | 6 |
| Ga | 0.411 | 2.860 | 7 |
| Al | 0.277 | 4.057 | 15 |
| Ni | 0.098 | 1.047 | 11 |
| Au | 0.061 | 0.921 (0.086 *) | 15 (1.4 *) |
| Ti | 0.053 | 3.539 | 67 |
| Cu | 0.793 | 17.760 | 22 |
| W | 0.022 | 0.573 | 26 |

The production of one 200 mm wafer with GaN on Si technology at CEA LETI requires a total of 655 L of ultra-pure or deionized water (UPW), 286 kWh of electricity and 61 kg of materials for the processing. It includes different processing gases such as $N_2$ (21.5 kg), a number of different acids (nitric (14.6 kg), hydrofluoric (0.1 kg), hydrochloric (2.6 kg) and sulfuric (1.7 kg)), bases, oxidizing agents (peroxide (4.7 kg)), other reactive chemistries and compounds such as CMP slurries (8.6 kg), solvents (isopropanol (2.0 kg)) or stripping and resist chemicals. The production of one wafer with GaN on Si technology at CEA LETI also

consume large volumes of water and energy for the facilities: 9.8 m³ of process cooling water, 342 kWh of electricity and 2.8 kg of natural gas. The data presented in this study are summarized and compared with inventory data reported in literature for semiconductor production in Table 3.

The life cycle inventory for processing of MOSc-HEMT technology is within the same order of magnitude as the life cycle inventory for production of other integrated circuits such as CMOS logic. The exact values can vary depending on the wafer size, technology, date, variations in recipes, number of mask levels, number and type of steps, type of fab (R&D or industry) and methodology for data collection.

After normalization and ponderation, the following impact categories are selected: resource use, fossils; climate change; resource use, minerals and metals; and ionizing radiation, accounting for 81% of the unique score (total of 55.7 mPt). The main environmental impacts for the production of a wafer are presented in Figure 3.

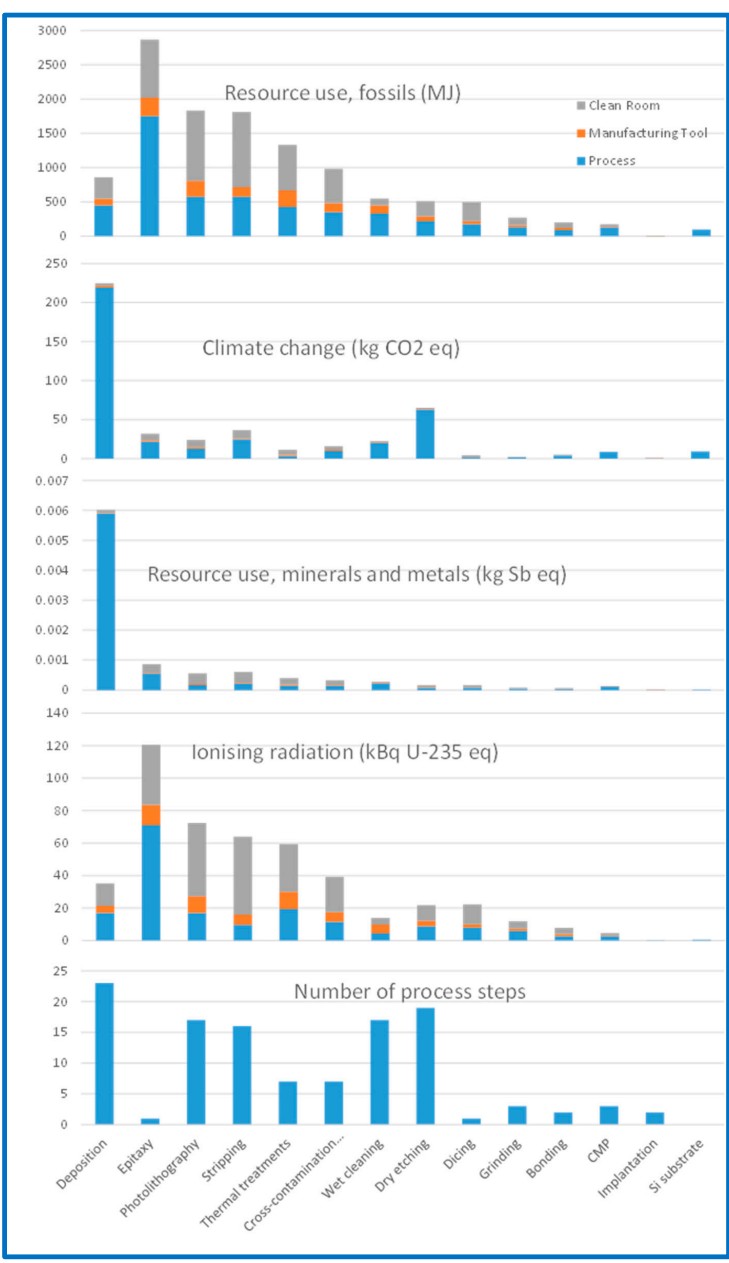

**Figure 3.** Main environmental impacts and number of process steps for the processing of a 200 mm GaN/Si MOSc-HEMT wafer presented by process category and by contribution (process, manufacturing tool and clean room.

Table 3. Amount of water, electricity and chemicals required for the processing of one cm$^2$ of wafer. * Data estimated with cumulative energy demand (CED) approach [2].

| Water (L/cm$^2$) | Electricity (kWh/cm$^2$) | Chemicals (g/cm$^2$) | Wafer Diameter | Data Year | Technology, Reference |
|---|---|---|---|---|---|
| 2.1 UPW + 31.2 = 33.3 | 0.91 processing + 1.09 facilities = 2.00 | 194, including: N$_2$ 68 Slurry 27 H$_2$SO$_4$ 5 H$_2$O$_2$ 15 Isopropanol 6 | 200 mm | 2021 | GaN/Si, from substrate to backside metallization, this study |
| | 2.43 * | | 3 inch | | GaN/Si, substrate + front-end [2] |
| | 1.94 processing + 1.08 facilities = 3.02 | N$_2$ 204 Slurry 9.4 Isopropanol 3.1 | 300 mm | 2005 | Si CMOS [25] |
| | 0.55 processing + 0.35 facilities = 0.91 | | 300 mm | 2002–2006 | Si CMOS, 130 nm technology node [14] |
| 6.3 | 0.67 | 7.3 | Average | 2011 | ST Microelectronics Rousset foundry [12] |
| | | N$_2$ 9.4 H$_2$SO$_4$ 61.8 H$_2$O$_2$ 21.9 | 300 mm | 2013 | ST Microelectronics Crolles, CMOS 45 nm technology node (from Tables 4.13 and 4.14) [26] |
| 5 to 107 | | | All | 2010–2020 | Compilation of data from scientific literature, databases, foundry report and industry roadmap, 3 to 350 nm technology node [27] |

### 3.2. Impact Assessment for the Production of One 200 mm Wafer, Processed with GaN on Si MOSc-HEMT Technology

The main contributor to resource use, fossils (total of $1.19 \times 10^4$ MJ) is the electricity consumption ($9 \times 10^3$ MJ), due to the use of uranium fuel for nuclear energy in the French electricity mix. For the same reason, electricity consumption also contributed significantly to ionizing radiation (467 out of a total of 469 kBq U 235 eq). Figure 3 shows that both contributions from the processing (including the eventual abatement systems) and the clean room facilities have high impacts on the resource use, fossils and ionizing radiation impact categories, as electricity is the main contributor to these impacts. Schmidt et al. also reported that environmental damage for DRAM and logic wafer fabrication is dominated by energy consumption, with ~65% of the electricity consumption for the processing [25]. Krishnan et al. reported 61% of the electricity consumption for the processing [14]. In our study, processing accounts for an average of ~46% of the electricity consumption (see Table 3). Most of the electricity consumption comes from the clean room infrastructure, allocated to each process step according to the ratio of tool surface/clean room surface and wafer-second processing time. Therefore, process categories with long duration processing steps and large tools appear with high clean room energy consumption. Concerning energy consumption of the processes, the epitaxy step is especially energy-intensive due to high temperature, long duration and presence of an abatement system (direct electricity measurement, see Table 1). Since process energy demand for plasma and ion generation and electrical heating may vary depending on the recipe (RF power, temperature) [28], electricity consumption measurements for each process step could help a more accurate assessment of the environmental impacts and identification of the hotspots. Also, for the epitaxy, consumption of ammonia for the processing accounted for 212 MJ.

The main contributors to climate change (total of 457 kg $CO_2$ eq) were emissions to air coming from two tools which were not equipped with an abatement system, and the processes involved fluorinated compounds such as $C_2F_6$ and $NF_3$ (PECVD) or $C_4F_8$ and $CHF_3$ (dry etching). Another dry-etching tool involving $SF_6$ and $NF_3$ was equipped with an abatement system and therefore did not contribute significantly to climate change. Other contributors included electricity consumption, natural gas consumption of the abatement systems and production of hydrogen and chemicals (EKC265, nitric acid, ammonia). Schmidt et al. reported that the impacts on climate change for DRAM and logic wafer fabrication are dominated by energy consumption, with ~75% of the greenhouse gases emissions, and only 6% from direct emissions of greenhouse gases such as FCs [25]. In our study, the portion of impact on climate change related to direct emissions of greenhouse gases is much higher.

The main contributors to resource use, minerals and metals (total of 9.6 g Sb eq) is one gold physical vapor deposition (PVD) step (5.79 g Sb eq), with the use of 0.086 g of "Gold {GLO}| market for | Cut-off, U". Gold data in the ecoinvent database are especially impactful due to the high impacts from gold production around the world. In the literature, it has already been reported that gold is among the metals displaying the highest environmental burdens [29] and rock-to-metal ratio [24]. It should be noted that only 0.061 g were actually deposited on the wafer for backside metallization. We estimated that 0.921 g of gold are consumed for this deposition step, i.e., 15 times the amount of gold effectively deposited on the wafer, and that 0.834 g of gold were recovered from gold deposited on the metal PVD target and the PVD chamber; therefore, only 0.086 g of gold were consumed for the deposition of 0.061 g on the wafer. The transportation of the metal PVD target and PVD chamber to the recovery plants, and the processes of cleaning, purification and valorization, were not taken into account and could lead to an increase in the impacts. It appears critical both to reduce the use of gold and to increase the recycling of gold in semiconductor manufacturing. Electricity contributes to 3.09 g Sb eq, mainly due to the use of copper for the distribution and transmission network. Gallium and silicon, which are present in higher amounts than gold (see Table 2), do not contribute much to resource use, minerals and metals. However, LCA does not take into account economic or

geopolitical aspects that can hinder the deployment of a technology or generate adverse impacts. For gallium, the estimated reserves are high but the annual production is low (550 t for 2022 according to USGS [30]). Ga is considered since 2011 as a critical raw material by the European Commission and its supply risk has increased in the last 2023 report [31].

### 3.3. Sensitivity Analysis for the Production of One 200 mm Wafer GaN on Si MOSc-HEMT

The impacts are calculated for a scenario with no gold recovery during the PVD deposition step (i.e., consumption of 0.931 g of gold). After normalization and ponderation, the following impact categories are selected: resource use, minerals and metals; resource use, fossils; and climate change, accounting for 82% of the unique score (total of 128 mPt, see Table 4). Gold material is typically used to reduce the electrical resistance, and, in some cases, it might be replaced easily by other metals. Here, since gold is used for the backside metallization, there is no strict requirement for a low resistance, and therefore gold could be replaced by another die-attach material, for instance by silver or aluminum, depending on the chosen packaging technology. The impacts are also calculated for a scenario with no use of gold at all (i.e., consumption of 0 g of gold). After normalization and ponderation, the following impact categories are selected: resource use, fossils; climate change; ionizing radiation; resource use, minerals and metals; and photochemical ozone formation, accounting for 83% of the unique score (total of 48.2 mPt). A variation in the amount of gold used in the processing leads to a large variation in the impacts on resource use, minerals and metals, here from +563% to −58%, as illustrated in Figure 4. Due to high impacts of gold material, the consumption of gold for semiconductor manufacturing should be calculated accurately for representative impact assessment. In case of replacement by another metal, the environmental impacts should be calculated with adjusted metal thickness to obtain the desired resistance.

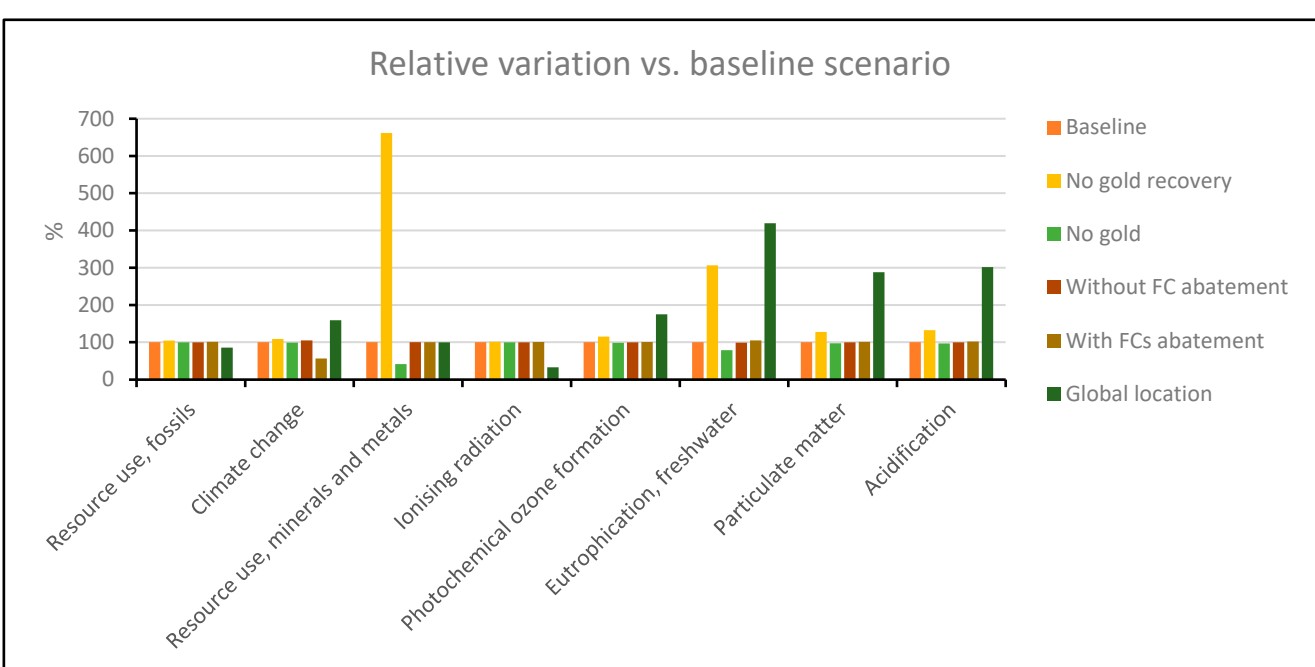

**Figure 4.** Relative variation of main impact categories according to different 200 mm MOSc-HEMT wafer production scenarios with variation from the baseline.

In the scenario with no gold consumption, since the impacts associated with resource use, minerals and metals are reduced, the impacts on photochemical ozone formation become apparent, even if it does not increase from the baseline scenario. The main contribution is the solvent waste (1.15 kg NMVOC eq out of 1.81 kg NMVOC eq) due to the emission of volatile organics into air.

**Table 4.** Impact assessment characterization values and the unique score obtained after ponderation for different 200 mm MOSc-HEMT wafer production scenario, with variation from the baseline. The characterization values for impact categories accounting for 80% of the single score are highlighted in bold.

| Impact Category | Unit | Baseline Scenario | No Gold Recovery | No Gold | Without FCs Abatement | With FCs Abatement | Geographical Location: Global |
|---|---|---|---|---|---|---|---|
| Single score | mPt | 55.7 | 128 | 48.2 | 56.2 | 50.6 | 71.6 |
| Resource use, fossils | MJ | **$1.19 \times 10^4$** | **$1.24 \times 10^4$** | **$1.18 \times 10^4$** | **$1.19 \times 10^4$** | **$1.20 \times 10^4$** | **$1.02 \times 10^4$** |
| Climate change | kg $CO_2$ eq | **457** | **499** | **455** | **480 (+5%)** | **259 (−43%)** | **730 (+60%)** |
| Resource use, minerals and metals | g Sb eq | **9.6** | **63.6 (+563%)** | **4.0 (−58%)** | **9.6** | **9.6** | **9.6** |
| Ionizing radiation | kBq U235 eq | **469** | 474 | **469** | 468 | 473 | 154 (−67%) |
| Photochemical ozone formation | kg NMVOC eq | 1.8 | 2.1 | **1.8** | 1.8 | **1.8** | 3.2 |
| Eutrophication, freshwater | kg P eq | 0.071 | 0.218 | 0.056 | 0.070 | 0.074 | **0.298 (+319%)** |
| Particulate matter | disease inc. | $1.02 \times 10^{-5}$ | $1.31 \times 10^{-5}$ | $0.99 \times 10^{-5}$ | $1.02 \times 10^{-5}$ | $1.03 \times 10^{-5}$ | **$2.94 \times 10^{-5}$ (+188%)** |
| Acidification | mol $H^+$ eq | 1.16 | 1.54 | 1.12 | 1.15 | 1.18 | **3.50 (+202%)** |
| Water use | $m^3$ depriv. | 185 | 196 | 184 | 183 | 197 | 267 |
| Eutrophication, terrestrial | mol N eq | 2.44 | 3.57 | 2.33 | 2.44 | 2.48 | 7.08 |
| Ecotoxicity, freshwater | CTue | $2.92 \times 10^3$ | $6.63 \times 10^3$ | $2.53 \times 10^3$ | $2.91 \times 10^3$ | $2.93 \times 10^3$ | $3.96 \times 10^3$ |
| Eutrophication, marine | kg N eq | 0.376 | 0.472 | 0.366 | 0.373 | 0.383 | 0.820 |
| Human toxicity, non-cancer | CTUh | $5.1 \times 10^{-6}$ | $8.1 \times 10^{-6}$ | $4.8 \times 10^{-6}$ | $5.1 \times 10^{-6}$ | $5.2 \times 10^{-6}$ | $8.7 \times 10^{-6}$ |
| Human toxicity, cancer | CTUh | $1.6 \times 10^{-7}$ | $2.1 \times 10^{-7}$ | $1.6 \times 10^{-7}$ | $1.6 \times 10^{-7}$ | $1.7 \times 10^{-7}$ | $2.5 \times 10^{-7}$ |
| Land use | Pt | $1.18 \times 10^3$ | $1.57 \times 10^3$ | $1.13 \times 10^3$ | $1.17 \times 10^3$ | $1.19 \times 10^3$ | $1.94 \times 10^3$ |
| Ozone depletion | kg CFC11 eq | $2.7 \times 10^{-5}$ | $2.8 \times 10^{-5}$ | $2.7 \times 10^{-5}$ | $2.7 \times 10^{-5}$ | $2.7 \times 10^{-5}$ | $2.8 \times 10^{-5}$ |

The impacts of the choice to abate fluorinated compounds are evaluated. FC emissions into air are calculated using destruction removal efficiency (DRE) abatement rates, and additional consumptions (water, natural gas, electricity, $O_2$) of the abatement systems are added for the seven PECVD, one CVD and two dry-etching steps using FCs in the process recipe, not already abated. After normalization and ponderation, the following impact categories are selected: resource use, fossils; resource use, minerals and metals; climate change; ionizing radiation; and photochemical ozone formation, accounting for 83% of the unique score (total of 50.6 mPt, (see Table 4). Since the impacts associated with climate change are reduced, the impacts on photochemical ozone formation become apparent on single score results, even if it does not increase from the baseline scenario. FC emissions into air are also calculated for a scenario without any abatement system for the three dry-etching process steps using FCs in the process recipe, already abated in the baseline scenario. In this case, the main indicators are resource use, fossils; climate change; resource use, minerals and metals; and ionizing radiation, accounting for 81% of the unique score (total of 56.2 mPt). Since fluorinated compounds have a high impact on climate change, taking into consideration the absence or presence of abatement can lead to an increase of 5% or a decrease of 43% of the impacts (see Figure 4). Therefore, it is important to properly assess the amount of fluorinated compounds used in the processes, the formation of by-products and the destruction removal efficiencies.

In order to evaluate the impact of the geographical location, the impacts are assessed using more global datasets from the ecoinvent database: in order of preference market for global {GLO}, and when not available rest of world {RoW}. A global electricity mix is used instead of French electricity mix. After normalization and ponderation, the following impact categories are selected: climate change; resource use, fossils; resource use, minerals and metals; eutrophication, freshwater; particulate matter; and acidification, accounting for 81% of the unique score (total of 71.6 mPt, see Table 4). The most notable differences, illustrated in Figure 4, are an increase of 60% in the impacts on climate change, with a total of 730 kg $CO_2$ eq instead of 457 kg $CO_2$ eq, and a decrease of 67% in the impacts on ionizing radiation, due to the use of coal and natural gas for electricity generation worldwide. Impacts on eutrophication, freshwater; particulate matter; and acidification also increase (+319%, +188% and +202%, respectively), mainly due to the contribution from the electricity mix. Indeed, electricity consumption alone contributes to 537 kg $CO_2$ eq, 0.248 kg P eq, $2.29 \times 10^{-5}$ disease inc. and 3.47 mol H+ eq due to coal mine operation, treatment of spoil from lignite mining and hard coil mining.

These results allow for the identification of the most efficient measures to reduce the environmental impacts. Gold should be, at a minimum, recovered from the processing, and substitution materials should be investigated. Studies can be carried out in order to reduce the consumption of, or to substitute fluorinated gases in, processing recipes. Gas abatement systems can be installed to cover 100% of process-gas emissions for existing and new tools, and their design can be improved. Finally, since electricity consumption is a major contributor to the environmental impacts, efforts should be taken into reducing both processing-related energy consumption and facility-related energy consumption. For instance, the latter can be reduced though various measures, such as exclusive sourcing and use of energy from low-carbon renewable sources, greater energy efficiency of buildings, reducing air pressure, increasing humidity, limiting air exchange in unused areas or eliminating leaks in air supply lines.

### 3.4. Impact Assessment of a Bare Die; Sensitivity on Die Size and Yield

In R&D, only hundreds of dies are present on each wafer (due to the presence of R&D-specific test structures on the mask). Also, for a 650 V 30 A MOSc-HEMT transistor ($R_{ON}$ of 54 m$\Omega$), given that the R&D die size after dicing is 9.49 mm$^2$, and that the die area on the mask is 9.98 mm$^2$ (a 150 μm margin between two dies is needed for the dicing), we consider that a maximum of 2782 dies with the mentioned specifications can be produced on a 200 mm wafer (with 6 mm edge exclusion), with 100% yield.

Further improvement of the 650 V 30 A MOSc-HEMT design with implementation of drain and source pads on top of the active area, as usually practiced in industrial manufacturing, would lead to a reduction of die area on the mask to 4.19 mm$^2$ (die size after dicing of 3.82 mm$^2$), allowing for a maximum of 6628 dies (100% yield). Expert estimations for the overall wafer processing yield for power devices is 75% for SiC and 90% for Si [23]. Adjusting the impacts of the production of a 650 V 30 A MOSc-HEMT die with yield varying from 75 to 90% leads to a maximum production of 4971 (75% yield) to 5965 (90% yield) dies with the mentioned specifications on a 200 mm wafer (with 6 mm edge exclusion). The environmental impacts of a MOSc-HEMT with different ON-current specification, similar design and same blocking voltage can be calculated by linearly adjusting the die size (e.g., ×2 for 60 A).

The environmental impacts of the production of a bare die can be calculated by dividing the impacts of a 200 mm wafer (Table 4) by the number of dies. For instance, the impact on climate change could vary from 0.04 kg $CO_2$ eq (scenario with FC abatement, production in France, 90% yield, 3.82 mm$^2$ die) to 0.15 kg $CO_2$ eq (scenario with global production, 75% yield, 3.82 mm$^2$ die). Environmental impacts can be triple-folded when considering a R&D design with larger die size: 0.35 kg $CO_2$ eq (scenario with global production, 75% yield, 9.49 mm$^2$ die).

The size of the package in this study is 8 × 8 mm$^2$; therefore, the die-to-package ratio then varies from 6.7 to 16.7, depending on the design (R&D- or industry-optimized). It should be noted that DFN 5 × 6 mm$^2$ packages are also commercially available and can be appropriate for such dies.

### 3.5. Life Cycle Inventory for the Production of the Packaging

Once the wafer is diced, the dies are packaged into chips. The material amount present on an 8 × 8 mm$^2$ DFN packaging is shown in Table 5. It is an underestimation of the total material amount necessary for the processing; the losses in materials during processing have not been taken into account. The DFN packaging is standard, with the exception of the interconnections between the source, drain, gate of the die and the Cu/Ni lead frame, which are realized by gold wire bonding. Gold bonding wires are not commonly used anymore due to the high gold price but can be found on some commercial components, usually in order to reduce electrical resistance (e.g., gold accounts for 5.14 mg out of a total chip weight of 2.093 g for the component Infineon IGOT60R070D1 600 V CoolGaN™ enhancement-mode Power Transistor [32]). Gold can also be found in the packaging as a very thin layer for plating contact pads [9]. The die-attach on the backside is performed with a silver-based solder paste, but the amount of materials was not included in this study due to lack of data.

**Table 5.** Amount of various elements present on the 8 × 8 mm$^2$ DFN chip packaging with gold wire bonding.

| Element | Calculated Amount Present on the 8 × 8 mm$^2$ Package (mg) |
|---------|---------------------------------------------------------|
| Ni | 0.03 |
| Au | 12.12 |
| Cu | 37.40 |
| epoxy | 103 |

### 3.6. Impact Assessment of the Packaging

After normalization and ponderation, the following impact category is selected: resource use, minerals and metals, accounting for 89% of the unique score (total of 1.04 mPt), due to the use of gold (0.78 g Sb eq). Impact on climate change is 0.588 kg $CO_2$ eq, also due to the use of gold. Although in previous studies, back-end processing has not been a major focus of these life cycle studies due to lesser contribution to cumulative energy demand [2], the impact of gold material shows that it is necessary to include back-end in

the assessment of environmental impacts. It has already been suggested to replace gold wire with copper wires, which exhibit high electric conductivity and thermal properties. Kuo et al. calculated a reduction in environmental damage single score (Pt) by 2972 times by replacing gold wire by copper wire in lead frame packaging [33]. In this study, with replacement of Au by the same amount of Al or Cu, the impact on resource use, minerals and metals can be reduced by several orders of magnitude. However, this aspect needs to be studied by adjusting the metal quantity to obtain the desired resistance, reliability and thermal management. In addition, this impact assessment of packaging does not take into account the processes, emissions, and energy consumption and does not reflect an accurate estimation of the environmental impacts.

Furthermore, since GaN power devices have a much smaller area than their Si counterparts, it raises thermal design challenges (higher temperatures and difficulties to extract the heat). Therefore, the packaging of GaN power devices, both thermal management and PCB layout, should be adapted to have a good thermal transfer capability [8]. Other packaging solutions such as flip chip (FC) or ball grid array (BGA), a typical type of surface-mount package (SMP) can provide improved heat dissipation, and hence, impact the device lifetime. For instance, an ecological comparison of soldering and sintering die-attach technologies shows that sintering has lower impacts when considering increased lifetime [34]. Therefore, further studies with detailed packaging life cycle inventories for different packaging solutions are necessary in order to assess the environmental impacts of a packaged GaN die. These environmental impacts should be compared in regard to the packaged die lifetime.

## 4. Discussion

It has been reported that Si chip manufacturing (IGBT and diodes, total chip area 9 $cm^2$) contributes significantly to the environmental impacts of power module manufacturing, which is itself an environmental hotspot in a 150 kW traction inverter [10]. The main reported environmental impacts are on climate change, ozone depletion, ionizing radiation and fossil material consumption [10]. Since WBG power semiconductors are able to fulfill the same function with a smaller die, one can expect a reduction of the environmental impacts compared with Si. However, for SiC semiconductors, additional processing steps are needed to obtain the SiC powder and to grow SiC wafers [23]. For GaN/Si semiconductors, epitaxy is necessary in order to grow the III-N layers.

### 4.1. Comparison of GaN/Si with Si and SiC

In Table 6, the environmental impacts for the production of a GaN/Si MOSc-HEMT are compared with publicly available results for Si and SiC power devices.

**Table 6.** Publicly reported environmental impacts for the production of a power transistor die (=bare die) or chip (=packaged die). * Data from this study.

| Technology | | Electrical Specifications | Die or Chip Size ($mm^2$) | Electricity Mix | Kg $CO_2$ Eq |
|---|---|---|---|---|---|
| GaN MOSc-HEMT * | Die | 650 V 30 A | 3.82 (90% yield) | France | 0.04 |
| GaN MOSc-HEMT * | Die | 650 V 30 A | 3.82 (75% yield) | Global | 0.15 |
| DFN package only with gold wires * | Chip | | 64 | n/a | 0.59 |
| GaN Fast Power IC 2020 [35] | Chip | | | | 0.2 |
| SiC MOSFET [3] | Die | 1200 V 150 A | 10.5 | Global | 0.9 |
| SiC MOSFET [36] | Chip | | | | <2.1 |
| SiC MOSFET [37] | Chip | 400 V | | USA | 250 |
| Si IGBT [3] | Die | 1200 V 75 A | 121 | Global | 3.3 |
| Si IGBT [37] | Chip | 400 V | | USA | 68 |
| Si IGBT [38] | Chip | | | China | 16.7 |
| Si FET 2020 [35] | Chip | | | | 0.7 |

It is clear that publicly available LCA information of sufficient level of detail and quality is limited, especially when communicated by manufacturers, as limited information on electrical specifications or die size are available. Musil et al. [3] assessed the climate change environmental impacts for the production of Si and SiC dies by extrapolation of energy requirements for Si wafers and Si wafer processing from the ecoinvent database and adding energy requirements for SiC wafer production [23]. Li et al. assessed the climate change environmental impacts for the production of Si, taking into account the mass of the chip (assimilated to silicon) and electricity requirements for front-end processing [38]. The Navitas study reports the climate change impacts of WBG being up to four times lower compared with a Si power semiconductor (Si FET) with today's production, and could be 10 times lower in the future as wafers of larger diameters are fabricated [35]. At this stage, it is not possible to conclude on potential improved or reduced environmental impacts for the manufacturing of WBG compared to Si power devices, even if it seems that reducing the die size has a great effect on the reduction of the impacts.

*4.2. End-Use Application*

The packaged chips or modules are incorporated into different end-use applications. The key features of WBG devices play an important role. The higher switching frequencies and the lower energy conversion losses allow the design of compact and lightweight end products with WBG semiconductors, consequently also using less materials. The extent and relevance of this effect depends on the end-use application and must be investigated at the product level to understand how they translate into energy and environmental impacts. Some studies indicate that reduction of $CO_2$ emissions at manufacturing could be achieved thanks to the use of WBG for USB-C chargers [35], but this effect can be counterbalanced by added functionality, such as the addition of a separate USB-A charging output, leading to rebound effects [39]. The relative size of the environmental impacts related to WBG semiconductor chips is expected to increase with the reduction in size of the other parts of the converters, in comparison with Si devices, highlighting even more the need for accurate WBG chip environmental assessment.

*4.3. Use Phase*

The use phase of semiconductor devices results in indirect environmental and health impacts resulting from energy-related emissions. Depending on the electricity mix in the location of use, the use phase may be the largest contributor among its life cycle stages for impact categories related to energy generation. In case of CMOS-based logic devices, the use phase has been shown to dominate impacts over the product life cycle, especially for smaller technology nodes (USA, 1995–2010 data [14,22]). In the field of semiconductors for ICT, the majority of environmental impacts are associated with the manufacturing stage in France [40]. For power electronics, the majority of the environmental impact is assumed to be due to the use phase, because power converters could be considered use-intensive devices, no matter the location. Many studies have quantified the energy gains enabled by the introduction of WBG semiconductors, gains which vary widely depending on the end-use application. Key variables in this case are the energy efficiency gains, the use intensity, and the product lifetime. For PV inverters, average energy efficiency gains of 2% are estimated, while for consumer electronics, a range of between 3 and 9% of energy efficiency gains is expected [1]. For electric vehicle applications, the overall contribution of the manufacturing phase of wide band gap power electronics to the total life cycle is quite small in terms of energy compared to the use phase consumption [2]. The ESOI (Energy Saved On energy Invested) metric shows that, despite estimated higher energy requirement for the manufacturing of SiC MOSFET vs. Si IGBT, the ESOI is favorable to SiC-based inverters for electric vehicles and photovoltaic inverters [37]. A study by Glaser et al. shows that for USB-C charging applications, whether WBG chargers have less impact on climate change than Si chargers depends on the number of charging cycles (Austria) [39].

The use phase stage should be defined for each application with realistic load and charging scenario, the location, frequency and length of use, and market adoption projections.

Finally, the environmental impacts on resource use, fossils and, to some extent, on climate change and on ionizing radiation, are related with energy consumption; therefore, one can expect that for use intensive power converters, the use phase is predominant for these impact categories. However, consumption of metals such as gold, emissions of greenhouse gases such as fluorinated compounds, and emissions of volatile organics during the semiconductor manufacturing stage can also lead to several impacts on the environment, especially on resource use, minerals and metals; climate change and photochemical ozone formation should be considered in order to assess the environmental impacts on the total life cycle.

### 4.4. End-of-Life

End-of-life impacts are not included in this study. After 2006 EU's Restriction on Hazardous Substances (RoHS), most manufacturers have switched to lead-free solders, avoiding lead emissions at end-of-life. While other effects from end-of-life disposal may exist, they have never been specifically measured and therefore are not included in this discussion [22]. However, electronic waste has become an urgent problem worldwide since it is more complex in composition than ordinary solid waste. The process of treating and disposing electronic waste should be considered, with a preference for reuse over material recovery in order to reduce costs, further consumption of energy and resources, and to mitigate potential environmental pollution. This seems especially true for dies whose manufacturing is energy- and material-intensive.

### 5. Conclusions

For the first time, a detailed cradle-to-gate life cycle assessment has been carried out for the manufacturing of a GaN/Si power device (MOSc-HEMT technology on 200 mm Si wafers), based on in-house CEA-LETI R&D microelectronics clean rooms.

Semiconductor wafer processing is material- and energy-intensive, with between 6 and 67 times more metals needed for the processing than present on the wafer, and consumptions of 194 g of chemicals (gases and liquids), 33 L of water and 2 kWh of electricity per $cm^2$ of processed wafer. High energy consumption of the processes and of the clean room facilities leads to environmental impacts on resource use, fossils; ionizing radiation; and climate change. Due to high electricity consumptions, these impacts are dependent on the electricity mix at the manufacturing location and may vary. Direct emissions of greenhouses gases such as fluorinated compounds during deposition or dry-etching processes have a high impact on climate change, which can be reduced with appropriate gas-abatement systems. The use of volatile solvents leads to impacts on photochemical ozone formation. The eventual presence of gold metallization or wire bonding in the wafer processing or in the packaging leads to high impacts on resource use, minerals and metals. Gold should be, at a minimum, recovered from the processing, and substitution materials should be preferred.

The life cycle inventory is not exhaustive, and this life cycle assessment accuracy can be further improved by taking into account the consumption of Si test wafers, purity grade of chemicals, direct electricity consumption measurements, actual industrial throughput, die size and yield and packaging processes.

Future work is needed in order to take into account the contribution of use and end-of-life stages. In order to examine the effectiveness in reducing the environmental impacts associated with a new power technology compared to another power technology, it is necessary to carry out a study at the converter product level for a detailed application and functionality (location, use intensity, product lifetime) and technology deployment projections.

**Author Contributions:** The manuscript was written through contributions of all authors. Investigation: G.G.; Data curation and Methodology: J.-C.L.B.; Writing—original draft, L.V. and G.G.; Writing—review & editing, L.V.; Formal analysis, L.V. and L.D.C.; Supervision, L.D.C. All authors have read and agreed to the published version of the manuscript.

**Funding:** This work was funded by the IPCEI French national program "Nano2022" and by the French ANR via Carnot institute.

**Institutional Review Board Statement:** Not applicable.

**Informed Consent Statement:** Not applicable.

**Data Availability Statement:** Data is contained within the article.

**Acknowledgments:** The authors want to acknowledge Messaoud Bedjaoui, Fanny Bertrand, Jérôme Biscarrat, Julien Bouchard, Thierry-Rene Braisaz, Matthew Charles, Pascal Chausse, Guillaume D'Alonzo, Thierry Enot, Gregory Enyedi, Virginie Enyedi, René Escoffier, Murielle Fayolle-Lecocq, Gennie Garnier, Charlotte Gillot, Josua Guerid, Romain Gwoziecki, Antonin Holo, Charley Lanneduc, Cyrille Le Royer, Blend Mohammad, Laurent Ortiz, Thomas Philippe, Patricia Pimenta-Barros, Rémi Riat, Yannick Rivoira, Léa Roulleau, Simon Ruel, Isabelle Servin, Richard Souil, Véronique Sousa, Rémi Torrecillas and all the other persons not mentioned here who contributed to the MOSc-HEMT process flow data collection, LCI data collection for processing in the clean room tools, establishment of the LCA methodology adapted for CEA-LETI clean rooms and fruitful discussions.

**Conflicts of Interest:** The authors declare no conflict of interest.

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
