# Peer review of "Cradle-to-Gate Life Cycle Assessment (LCA) of GaN Power Semiconductor Device"

_sustainability, doi:10.3390/su16020901_

Round 1

Reviewer 1 Report

Comments and Suggestions for Authors

The article is very interesting and aligns with the objectives of the journal. For its publication, this reviewer identifies the following recommendations:

1.      The authors should incorporate similar works from the literature, comparing what has been done so far and highlighting the novelty of this study. Justifying the scientific contribution of the article is crucial. This is mandatory.

2.      LCA is a methodology. Please remove any references to "technique" or "tool."

3.     The authors need to explain why they did not include all stages in this LCA and performed a "cradle-to-gate" analysis. This is not a criticism of the work but an establishment of analysis transparency.

4.      In the "goal and scope" section, authors should add study assumptions, such as whether all unit processes were included in this "cradle-to-gate" analysis, if any input was not considered, and if all inventory inputs were found exactly the same in the software database.

5.      In the "goal and scope" phase, indicate the methodology used (midpoint, endpoint) as well as the impact categories analyzed. Also, specify the databases used. EF database 3.1 is confused with the EF methodology. Please correct, clarify, and provide detailed explanations about the methodology employed and the database. This is mentioned in rows 56 and 212.

6.      Section 2.4, Sensitivity Analysis, lacks substantial information. It is necessary to expand on the justification and explain how it was conducted.

7.      Figure 3 is difficult to understand due to its representation. Kindly recreate the figure, separating the coordinate axes, indicating axis information, providing legends, ensuring consistency in font size/style, and specifying units. Currently, it appears to be a graph directly copied and pasted from Microsoft Excel.

8.      The lack of a proper literature review makes it impossible to contextualize these results. Please include a discussion section where the coherence of the objective results is established according to existing literature.

9.      Review the author guidelines. There are tables without titles, and the references do not follow the journal's style.

10.   The references are limited. The authors should conduct a thorough review of the state of the art. This reviewer performed a quick search and identified several interesting studies that should be cited in this work.

Reviewer 2 Report

Comments and Suggestions for Authors

The authors used life cycle assessment to evaluate data from cradle to grave during the GaN/Si power period. Environmental hot spots related to different materials and processes were identified, such as electricity consumption, greenhouse gases, direct emissions of volatile organic chemicals, etc. The findings support most of the conclusions and the manuscript can be considered for publication after minor revisions. Some minor comments are as follows:

1. It is recommended to use a three-line table for the table.

2. The title of section 3.3 is too long. It is recommended that the author revise it.

3. In order to better attract readers, it is recommended that the author add some relevant research conclusions to the abstract.

4. Some important literature on LCA should be referred to, such as: Science of The Total Environment 814 (2022) 152870; Energy, 284 (2023) 129315. 

5. Can the author provide a flow chart so that readers can better understand the content of this article?

Comments on the Quality of English Language

Moderate editing of English language required

Reviewer 3 Report

Comments and Suggestions for Authors The paper presents the results of a life cycle assessment (LCA) from cradle to gate for a GaN/Si power device, identifying environmental critical points associated with different materials and processes, such as electricity consumption, direct greenhouse gas emissions, use of gold, and volatile organic chemicals. The LCA methodology used in the study was EF 3.1, and the results were compared with publicly available data for Si, GaN, and SiC power devices. The study analyzed the environmental impacts of producing a GaN semiconductor power device, considering all phases from raw material extraction to the production gate. The LCA methodology followed the four stages of goal and scope definition, inventory analysis, impact assessment, and interpretation. The LCA study results provide information on energy and resource use during the manufacturing processes of GaN power semiconductor devices, contributing to a better understanding of their environmental performance.
Some limitations should be addressed before publication. For example, the article acknowledges the lack of available life cycle assessment (LCA) data for gallium nitride (GaN) and silicon carbide (SiC) power semiconductor materials, indicating a limitation in existing research on these materials. The study focuses on a cradle-to-gate LCA, meaning it assesses the environmental impacts of the GaN/Si power device from raw material extraction to the production gate. However, it does not consider the entire life cycle of the device, including its use and end-of-life stages. The article does not provide specific details on sample size or the methodology used for data collection, limiting the generalization of the results. The study compares results with publicly available data for Si, GaN, and SiC power devices. Still, it does not offer information on specific sources or the reliability of the data used for comparison. The article does not address potential limitations or uncertainties associated with the EF 3.1 LCA methodology, which could affect the accuracy and reliability of the results.
I will list some points that could be improved for the publication of the article:
1- The text requires revision.
2- Figures must be enlarged; some may be split for better reader visualization.
3- Table formatting needs improvement; some are cut off.
4- Authors should seek more references for similar works and add them to the introduction. This modification will significantly enhance the discussion throughout the text.
Despite the limitations, the article provides valuable information and should be accepted for publication after revisions. Comments on the Quality of English Language

Minor editing of English language required

Reviewer 4 Report

Comments and Suggestions for Authors

Manuscript ID: sustainability-2670231

Title: Cradle to Gate Life-Cycle Assessment (LCA) of GaN Power Semiconductor Device

The authors determined the environmental impact of a GaN/Si power device by using a cradle-to-gate life cycle assessment. In the estimation of the impacts assessment characterization values, the authors proposed several scenarios and the results were discussed, in order to find the environment-friendly scenario. The results were compared with those published for Si, GaN and SiC power devices.

The subject is interesting and the manuscript is well written. Consequently, I recommend the manuscript for publication.

A minor observation would be that the numbers and captions of tables are missing.

Reviewer 5 Report

Comments and Suggestions for Authors

This article considers the entire life cycle assessment process during Gallium Nitride power semiconductors and proposes the presents a cradle-to-gate life cycle assessment for a GaN/Si power device, and compared the public data of Si, gallium nitride and silicon carbide power devices, the research content is innovative. However there are still parts of the paper that deserve further improvement.

1.The abstract explains the overall research process, but there are fewer words discussing the advantages and innovations of the article. The advantages and innovation of the comparative results should be further reflected.

2.There is little introduction to the latest research results and related research sections in the introduction.

3.There is fewer words for comparison of the overall results in the discussion.It is recommended to add more.

4.Add titles to the charts in the article and unify the format.

5.There are too few references and some documents are relatively old.. It is recommended to add more.

6.The format of the paper needs to be adjusted. For example, should the first letters of the titles of 4.2 and 4.3 be capitalized?

Comments on the Quality of English Language

Minor editing of English language required.
